# Communication Planning for Cooperative Terrain-Based Underwater Localization

**DOI:** 10.3390/s21051675

**Published:** 2021-03-01

**Authors:** Jacob Anderson, Geoffrey A. Hollinger

**Affiliations:** Collaborative Robotics and Intelligent Systems Institute, Oregon State University, Corvallis, OR 97331, USA; geoff.hollinger@oregonstate.edu

**Keywords:** autonomous underwater vehicles, terrain-based navigation, acoustic communication

## Abstract

This paper presents a decentralized communication planning algorithm for cooperative terrain-based navigation (dec-TBN) with autonomous underwater vehicles. The proposed algorithm uses forward simulation to approximate the value of communicating at each time step. The simulations are used to build a directed acyclic graph that can be searched to provide a minimum cost communication schedule. Simulations and field trials are used to validate the algorithm. The simulations use a real-world bathymetry map from Lake Nighthorse, CO, and a sensor model derived from an Ocean Server Iver2 vehicle. The simulation results show that the algorithm finds a communication schedule that reduces communication bandwidth by 86% and improves robot localization by up to 27% compared to non-cooperative terrain-based navigation. Field trials were conducted in Foster Reservoir, OR, using two Riptide Autonomous Solutions micro-unmanned underwater vehicles. The vehicles collected GPS, altimeter, acoustic communications, and dead reckoning data while following paths on the surface of the reservoir. The data were used to evaluate the planning algorithm. In three of four missions, the planning algorithm improved dec-TBN localization while reducing acoustic communication bandwidth by 56%. In the fourth mission, dec-TBN performed better when using full communications bandwidth, but the communication policy for that mission maintained 86% of the localization accuracy while using 9% of the communications. These results indicate that the presented communication planning algorithm can maintain or improve dec-TBN accuracy while reducing the number of communications used for localization.

## 1. Introduction

Researchers, militaries, and marine companies are increasingly utilizing autonomous underwater vehicles (AUVs) to improve the efficiency of underwater data collection applications like gathering scientific data, marine geology, marine animal and ecosystem monitoring, surveillance, naval mine mitigation, pipeline and infrastructure inspection, and bathymetric surveying [1,2,3]. Many of these data collection missions require that the AUV is able to localize itself and sense its surroundings. In many robotics applications, sensors such as GPS, LIDAR, and cameras are used for localization and sensing; however, electromagnetic waves quickly attenuate in water, prohibiting the use of these instruments. Inertial sensors are available, but they are either cost prohibitive or have too much noise and drift for reliable localization. Acoustic is the primary mode of ranging and communication, but similarly tends to be either cost prohibitive or provide low information throughput. Due to these restrictions, terrain-based navigation (TBN) has become a leading means of self-contained underwater localization [4].

Vehicles using TBN for underwater localization are dependent on terrain information to improve their state estimation. Vehicles traveling over areas with distinctive terrain will be able to localize better than those traveling over areas of smooth terrain. Cooperative localization allows vehicles with better state estimation to aid other vehicles. A vehicle with accurate localization can transmit its location and covariance to other vehicles via an acoustic modem. A receiving vehicle can calculate its distance from the transmitting vehicle and use this distance measurement with the information provided by the transmitting vehicle to improve its own localization [5].

This work focuses on planning when each AUV communicates its localization information which is important for two reasons: (1) overlapping communication can cause interference, resulting in failed communication, and (2) acoustic modem bandwidth is extremely limited and often needed to transmit other information, such as scientific data.

This work presents a decentralized communication planning algorithm that determines an optimized communication policy for collaborative underwater TBN localization. The algorithm forward simulates a group of two or more AUVs following predetermined paths. At each time step a scenario in which the hosting AUV communicates is considered and then compared to scenarios in which the hosting AUV does not communicate. Each communication incurs a cost, and scenarios that result in poor localization are discarded. The resulting communication policy contains a minimum number of communications while limiting the uncertainty in each vehicle’s location. The proposed algorithm enables the AUVs to either conserve energy or transmit other types of data. To our knowledge, this is the first algorithm to non-myopically plan communication policies for AUVs using cooperative localization.

The field trials were conducted at Foster Reservoir near Sweet Home, Oregon, USA. Two riptide micro-unmanned underwater vehicles (uUUVs) were used to collect GPS, altimeter, acoustic communication and dead reckoning data. These data were then used to evaluate the performance of the proposed communication planning algorithm by evaluating dec-TBN under four communication regimes: no communication, full communication using all of the available acoustic data, the communication policy in which acoustic communications were only used if they were selected by the policy, and random communication policies of varying bandwidths. The results show that the communication policies were able to maintain the localization accuracy of the dec-TBN algorithm while reducing the number of communications by 57% or more. In three of the four trails, the communication planning algorithm also enabled dec-TBN to localize more accurately than standard TBN and dec-TBN using a full communication regime. The results from the field trials and the simulations indicate that the planning algorithm is able to find a competitive communication policy without trial and error.

## 2. Related Work

This work draws on the subjects of Terrain-Based Navigation and communication planning. The next two subsections look at work that has been done in those fields.

### 2.1. Terrain-Based Navigation

TBN is a technique that originated in 1980 with TERCOM [6] where a flying object, such as a missile, would compare its altimeter readings to a digital elevation map. Recent implementations of TBN have been centered around particle filters that compare a vehicle’s altimeter readings to a bathymetry map [4,7]. Particle filter TBN methods can provide accurate vehicle location in areas of significant bathymetric features but tend to diverge on smooth terrain.

Tan et al. developed a decentralised TBN (dec-TBN) algorithm [5] where multiple vehicles share their locations and covariances with each other. For this algorithm, each vehicle hosts its own particle filter and performs regular comparisons of its altimeter readings to a bathymetry map. The vehicles also transmit their particle filter’s estimated location and covariance to the other vehicles acoustically. The vehicles that receive the acoustic message estimate their distances from the transmitting vehicle based on the message’s one way time of flight. That distance, along with the transmitting vehicle’s location and covariance, are used to update the receiving vehicle’s TBN estimate. In this work, the vehicles take turns communicating one after another with no consideration given to the timing of communication [5].

Dec-TBN forms the foundation of our work. We use a similar dec-TBN formulation in which each vehicle hosts a particle filter that is informed by altimeter measurements and location data transmitted from other vehicles. Our work builds on dec-TBN by examining communication planning for these vehicles to see if localization accuracy can be retained while reducing the communication overhead.

### 2.2. Communication Planning

To date, the majority of communication planning for state-of-the-art distributed localization is focused on choosing what data to share with other robots. One approach is to use metadata from the robots’ pose graphs to identify individual scan lines or camera images that may contain loop closures [8,9,10]. Another approach is to design linear–quadratic regulators to control data flow [11]. These approaches are intended for terrestrial application where communication bandwidth and reliability are significantly better than underwater applications. In this case the robots are able to transfer large data sets to each other. The bandwidth available on an acoustic modem precluded these approaches.

A more applicable line of research is communication planning for multi-robot coordination. These methods focus on communicating the belief states of the robots for the purpose of deciding what actions they should take [12]. This is analogous to dec-TBN where the AUVs share their state estimations to help each other localize more accurately.

Williamson et al. applied information theoretics to communication planning for multi-robot cooperation by using KL divergence to quantify the reward of an agent’s communication [12]. Their approach then uses this approximation in formulating a decentralized partially observable Markov decision process (Dec-POMDP) to remove reasoning over the value of communication from the POMDP’s coordination model. Using a deterministic formula to approximate the value of each communication reduced the search space of the POMDP [12].

Unhelkar and Shah followed the idea of assigning value to communication and proposed a decentralized Markov decision process (Dec-MDP) with a reward function that maximized the expected reward for communication [13]. Marcotte et al. built on the aforementioned Dec-MDP and Dec-POMDP by factoring the planning problem so that each robot could plan independently. Subsequent, their algorithm scaled linearly with the number of robots. Additionally, Marcotte et al. forward-simulated the outcomes of message passing to determine the value of each communication. This approach has the added advantage of being able to determine what the message content should be [14]. Similarly, Barcis et al. developed an evaluation model that determines the value of certain types of data. However, rather than using a Markov decision process, Barcis et al. built their evaluation model using domain knowledge of the application [15].

Best et al. considered planning-aware communication [16]. In this work a decentralized planning algorithm is presented in which a group of robots is attempting to complete a task. While the planner is evaluating which actions a robot should perform, it tracks its uncertainty in what actions it expects the other robots to perform. Once the uncertainty of a particular robot exceeds a certain threshold it requests a planning update from that robot. To minimize communication, the algorithm constructs a directed acyclic graph representing the uncertainty in the robots’ actions and communication cost. While constructing the graph, every time a communication was requested from a robot the uncertainty in the robot’s actions reduced to zero. Subsequently, the base node only had to be evaluated once. The implementation of this directed acyclic graph provides a communication schedule that is optimal with respect to belief and results in an algorithm with polynomial run time complexity [16]. Additionally, finding a communication policy is now equivalent to a longest path search through a directed acyclic graph, which has linear time complexity [17].

The communication planning problem presented here builds on these works in multi-robot coordination. The communication problem is factored so that each robot can plan its own communication separately from the other robots. Forward simulation is used to approximate the value of each communication. The simulations are then used to build a directed acyclic graph in which the robot is required to communicate once the uncertainty in a robot’s location exceeds a certain threshold. The graph can then be searched for a minimum cost communication policy. Unlike the aforementioned works, which assumed that the robots have deterministic transition functions, the proposed communication planning applies these techniques to the inherently stochastic problem of TBN.

## 3. Methods and Materials

This section starts by looking at the problem of planning communications for a group of AUVs using dec-TBN to localize. It then presents a decentralized communication planning algorithm that builds on the dec-TBN algorithm. This section finishes by illustrating how the algorithm was tested through simulations, and field trials with two AUVs. The subsection on methods walks through the problem formulation and the algorithms used to plan the communications. The materials subsection describes the simulations and the AUVs used for the field trials.

### 3.1. Methods

This subsection will first examine the problem of communication planning for a group of AUVs using dec-TBN for localization. It will then present a communication planning algorithm that solves this problem. This section will also present the dec-TBN algorithm that is a part of the communication planning algorithm.

#### 3.1.1. Problem Formulation

We are interested in the problem of scheduling communication for a group of AUVs using dec-TBN to localize. The AUVs travel in a mapped environment taking altimeter readings as they move. Each AUV uses the TBN algorithm to estimate its location and the AUVs can communicate their location and uncertainty with each other. We want to plan a communication policy that uses minimal transmissions while limiting the uncertainty in the AUVs’ location.

Each AUV *r* is equipped with an altimeter, a depth sensor, an acoustic modem, and a digital bathymetry map of the environment. The AUVs are also provided the starting locations $X_{t=0}^{r}$, initial covariances $\Sigma_{t=0}^{r}$ and paths of all the other AUVs. The AUVs will localize with a dec-TBN algorithm. An individual instantiation of dec-TBN consists of a particle filter that tracks the hosting vehicle’s location. At each time step, the inputs to dec-TBN are the most recent vehicle control inputs and altimeter reading. If available, $X_{t}^{i}$, $\Sigma_{t}^{i}$, and the intra-vehicle range from another vehicle, i≠r, are also provided to the dec-TBN. The dec-TBN algorithm outputs the position estimate $X_{t}^{r}$ and covariance $\Sigma_{t}^{r}$ of the hosting vehicle.

The AUVs can communicate with each other via the acoustic modems. If all of the vehicles have synchronized clocks, and assuming isotropic water temperatures, the distance between the transmitting and receiving vehicles can be calculated using the communication’s one way time of flight (OWTF). To perform dec-TBN, the AUVs must take turns communicating their localization statistics. The receiving vehicles use the localization statistics and distance measurement from the transmitting vehicle to inform the next dec-TBN update.

The proposed decentralized planning algorithm generates a communication policy $\pi^{r}=\left\{\pi_{1}^{r},\pi_{2}^{r},...,\pi_{T}^{r}\right\}$ that indicates when the hosting vehicle should communicate its $X_{t}^{r}$ and $\Sigma_{t}^{r}$ within the planning horizon *T*. The variable $\pi^{r}$ is a binary sequence, $\pi_{t}^{r}=true$ indicates that the vehicle should communicate at time step *t* and $\pi_{t}^{r}=false$ indicates that it should not communicate. If a vehicle communicates at time *t*, the transmission includes the time of the transmission, $X_{t}^{r}$ and $\Sigma_{t}^{t}$. The communication is received by all the other AUVs and informs their next dec-TBN update.

The objective of the planning algorithm is to minimize the AUVs’ communication while maintaining a bound on localization accuracy. Each communication is given a unit cost since every localization message consists of the same volume of information: an x-y location of the vehicle, a two by two covariance matrix, and a time stamp. The modem itself may append more information depending on its design, but this usually a proprietary matter that the end user does not have control over. The cost of π for *N* robots is the sum of all communication, i.e.:(1)$$Cost\left(\pi \right)=\sum_{i=1}^{N}\sum_{t=1}^{T}1\left(\pi_{t}^{i}=true\right)$$

To ensure that the AUVs maintain a certain degree of accuracy, the communication planning algorithm is constrained by the estimated accuracy σ of the AUVs’ localization. For a group of *N* AUVs at time step *t*, $\sigma_{t}$ is defined as:(2)$$\sigma_{t}=\sum_{i=1}^{N}trace\left(\Sigma_{t}^{i}\right)$$

The proposed planning problem is formulated as a constraint-based optimization to find $\pi^{*}$ that has the lowest communication cost while maintaining σ under a user defined threshold $\sigma_{max}$:(3)$$\pi^{*}=\underset{\pi }{\mathrm{argmin}}\;Cost\left(\pi \right):\sigma_{t}<\sigma_{max}\;\forall \;T$$

#### 3.1.2. Communication Planning Algorithm

The proposed communication planning method is a decentralized planning algorithm that is intended for vehicles using dec-TBN. The intuition behind the algorithm is that localization information from a vehicle with an accurate state estimate can be used to improve the localization of other vehicles. It may not be advantageous for vehicles with relatively poor localization to transmit their information either. Additionally, the dec-TBN particle filters use randomly generated noise to disperse the particles. Modeling a particle filter’s response to a vehicle’s path is impractical. For these reasons the proposed communication planning algorithm involves simulating vehicles traveling through the environment and evaluating the effects of vehicle’s communication on the group’s localization.

The communication planning algorithm is a decentralized algorithm that is run independently on each vehicle. The algorithm builds a directed acyclic graph *G*. The graph nodes *N* represent the $\sigma_{t}$ of the AUVs and the edges represent the communication cost between nodes. The graph is built by forward simulating a set of vehicles as they follow predefined paths through an environment. Each leaf node hosts a simulated state of the vehicles which is instantiated as a set of particle filters; e.g., if planning for three vehicles, each leaf node would contain three particle filters. The leaf nodes are expanded by forward simulating the vehicles one time step and having them localize via dec-TBN on the bathymetry map. The leaf nodes are expanded twice, once with the hosting vehicle communicating its $X_{t}$ and $\Sigma_{t}$, and once without communicating, thereby creating two new leaf nodes. The edges between the parent node and the new leaf nodes are given weights of 1 and 0 for communicating and not communicating respectfully.

To reduce computational demands of the algorithm, it is assumed that the state resulting from the host vehicle communicating is the same for all leaf nodes. This is a similar assumption to Best et al. [16] and the resulting structure of the graph is demonstrated in Figure 1. $N_{t,0}$ represents the state in which the host vehicle has communicated at time step *t*. The rest of the nodes result from not communicating. Constructing the graph as a directed acyclic graph provides polynomial run time with complexity $O(BNT^{2})$ where *B* is the number of particles used in the particle filters. Additionally, finding a communication policy from the directed acyclic graph can be done with linear time complexity [17].

In practice, the planning algorithm holds the simulations for each leaf node *i* in a queue $Q_{i}$. At each time step the algorithm cycles through the queue and progresses each simulation one step forward. The simulations move the robots R along a path *P* using a motion model *M*. Then, they update the dec-TBN algorithm using the simulated vehicles’ locations on the bathymetry map to provide depth readings. If communication is indicated, the $X_{t}$ and $\Sigma_{t}$ of the hosting vehicle’s dec-TBN algorithm is provided to the other simulated vehicles’ dec-TBN updates.

At each time step all of the simulations in *Q* are forward simulated without communication. If $\sigma_{t}$ of the resulting simulation state is less than $\sigma_{max}$, then the resulting state is added to *Q* and a new node is appended to *G* with the value of $\sigma_{t}$. This node is connected to the leaf node representing the state that was just updated with an edge weight of 0. $Q_{0}$ holds the simulation for $N_{t,0}$ in which the hosting vehicle has just communicated. The last update performed at each time step is to forward simulate $Q_{0}$ with communication. The resulting state is added to *Q* as the next $Q_{0}$ and a corresponding node is added to *G* with the simulation’s value of $\sigma_{t}$. This node is connected to all of the leaf nodes with an edge weight of 1. Due to dec-TBN’s reliance on terrain information to improve localization, a static threshold value for $\sigma_{max}$ is difficult to determine. Instead, the threshold value is a product of $\sigma_{max}$ and $\sigma_{t}$ of $Q_{0}$. See Algorithm 1.
**Algorithm 1** Communication Planning Algorithm**Require:**${\left[X_{0},\Sigma_{0},M,P\right]}^{r},\forall r\in R,T,\sigma_{max}$
**Ensure:**π▹ Robot Communication Policy 1: **▹ Particle filter for each robot**
 2: $Q_{0}.sims\leftarrow$PF($(\left[X_{0},\Sigma_{0},M,P\right]$)${}^{r},\forall r\in R$▹ Queue 3: $Q_{0}.parent\leftarrow 0$
 4: $G.nodes_{0}\leftarrow 0$▹ Cost Graph 5: m=0
 6: **for**
t=1 to *T*
**do**
 7:    P=Q.parent▹ List of current leaf nodes 8:    **▹ Update $Q_{i}$ without communication *a***
 9:    **for**
i=|Q| to 1 **do**
 10:        $\eta \leftarrow PF\_Update(Q_{i},a=false)$
 11:        **if**
$\eta .\sigma <Q_{0}.\sigma \times \sigma_{max}$
**then**
 12:           m=m+1
 13:           $p=Q_{i}.parent$
 14:           $Q_{i+1}.sims\leftarrow \eta$
 15:           $Q_{i+1}.parent\leftarrow m$
 16:           $G.nodes_{m}\leftarrow \eta .\sigma$
 17:           $G.edges_{p,m}\leftarrow 0$
 18:    **▹ Update $Q_{i}$ with communication**
 19:    m=m+1
 20:    $Q_{0}.sim\leftarrow$PF_Update($N_{0,t-1},a=true$)${}^{r},\forall r\in R$
 21:    $Q_{0}.parent\leftarrow m$
 22:    $G.nodes_{m}\leftarrow Q_{0}.\sigma$
 23:    $G.edges_{P,m}\leftarrow 1$
 24: π←LowestCostPath(*G*)▹ Search over Graph

#### 3.1.3. Decentralized TBN Algorithm

Many modern TBN algorithms use particle filters to track the vehicle’s position on a digital elevation map or bathymetry map in the case of marine environments [18,19]. The dec-TBN algorithm used here utilizes an update step that incorporates range measurements to another vehicle with that vehicle’s localization information. First, a vehicle transmits its $X_{t}$ and $\Sigma_{t}$ to the other vehicles via an acoustic modem. A receiving vehicle calculates a range measurement *D* from the transmitting vehicle via the acoustic communication’s OWTF. *D* provides a measurement that adds information to the particle filter and is used with the location information from the transmitting vehicle in the next particle filter update.

The modified particle filter update propagates the particles via the vehicle’s speed *S*, heading θ, time between updates δt, and motion model *M*. Then the particle filter calculates the probability of each particle’s location given a depth measurement *z* from an altimeter. A probability density function $pdf_{alt}$ using the altimeter’s mean and standard deviation is used to compare the depth measurement to the expected depths Z from the bathymetry map. The probability of each particle’s location is also calculated by creating a multivariate normal distribution $pdf_{coms}$ using the location and covariance received from the other vehicle. The locations of the particles are then shifted towards the transmitting vehicle’s location by the distance calculated from the acoustic modem’s time of flight. The resulting particle locations are used to sample the aforementioned normal distribution. The probabilities resulting from the bathymetry measurement and the communication measurement are multiplied together with the previous particle weights to create the new particle weights. This process is illustrated in Algorithm 2.
**Algorithm 2** Particle Filter Update with a Received Communication 1: **function**
PF_Update($X_{0},T,S,\theta ,z,D,X^{r},\Sigma^{r}$)
 2:    $particles\leftarrow X_{0}$▹ Initialize particles 3:    w←1/|particles|▹ Initialize particle weights 4:    **for**
t=0 to *T*
**do**
 5:        particles←move_particles(S,θ,t,M)
 6:        Z←BathymetryMap(particles)
 7:        $w_{bathy}\leftarrow$pdf${}_{alt}$ ($z-Z,\mu_{alt},\sigma_{alt}$)
 8:        $w_{comms}\leftarrow$pdf${}_{coms}$($particles+D,X^{r},\Sigma^{r})$)
 9:        $w\leftarrow w\times w_{bathy}\times w_{comms}$
 10:        w←w/∑(w)
 11:    X←∑(weights×particles)
 12:    Σ←Covariance(particles)
    **return**
X,Σ


### 3.2. Materials

This subsection will look at the simulator used to test the communication planning algorithm and the AUVs used to collect data during field trials.

#### 3.2.1. Simulations

To evaluate the proposed communication planning algorithm, simulations were run to determine how well AUVs using dec-TBN could localize given the communication policy produced by the algorithm. The simulations leverage real-world data by using a bathymetry map of Lake Nighthorse near Durango, Colorado, USA. The bathymetry map was created by doing an extensive survey of the reservoir. Depth readings from the survey were corrected for temporal changes in the height of the reservoir. Then, the depths were combined into a digital elevation model via a sliding window Kalman filter (see Figure 2). Additionally, the depth readings from this survey were used to build a sensor model of the altimeter on an Ocean Server Iver2 AUV that was used for part of the data collection. This sensor model was used in the simulator to provide depth readings to the simulated vehicles.

Simulations involved two or more AUVs following predefined paths across the bathymetry map. Each AUV used its dec-TBN state estimate to compute control inputs. Gaussian white noise was introduced into the vehicle’s true movements to emulate the navigational errors that occur in real underwater vehicles. The sensor model derived from the Iver2’s altimeter was used to introduce noise into the depth readings used for the AUVs’ dec-TBN. Figure 2 shows the simulated paths of four AUVs. The blue stars are waypoints that the AUVs followed. The green lines are the actual paths that the AUVs traveled when using dec-TBN. The orange lines are the actual paths that the AUVs traveled when using dead reckoning. The AUVs traveled a little more than 1.5 km in these simulations.

The AUVs’ localization accuracy in the simulation is evaluated by comparing the AUVs’ state estimations to their true locations. The error $\delta_{t}^{r}$ for AUV *r* at time step *t* is calculated as:(4)$$\delta_{t}^{i}=∥X_{t}^{i}-{\hat{X}}_{t}^{i}∥$$

In this case, *X* is the true location of the AUV and $\hat{X}$ is the TBN estimate.

The joint error $\Delta_{t}$ of all *N* AUVs is:(5)$$\Delta_{t}=\sum_{i=1}^{i=N}\delta_{t}^{i}$$

To evaluate the communication planning algorithm, a number of simulations were run utilizing different communication regimes: no communication, full communication, communications planned by our algorithm, and varying bandwidth policies. For full communication, the AUVs took turns communicating at each time step. The varying bandwidth policies involved the vehicles communicating in evenly spaced blocks. Each block of communication involved all of the vehicles taking a turn to communicate. The blocks were spaced so that the total number of communications used equaled a percentage of the full communication.

Before starting a simulation, communication policies were generated for the AUVs. For simulations using the planning algorithm, each AUV was given the initial locations of all the vehicles with a corresponding uncertainty of three meters. The AUVs were also given paths that each vehicle would follow. The AUVs then performed the communication planning algorithm independently. The AUVs did not share their communication policies, so it was possible that more than one vehicle tried to communicate at the same time. In this case, the communications were assumed to interfere with each other and were not received by any of the vehicles.

The simulation started after the communication policies had been generated. During the simulation, the AUVs attempted to follow their prescribed paths by using dec-TBN and communicating their locations and covariances at the time steps indicated by the communication policy. The simulation ended once the first AUV achieved its goal. The particle filter in the dec-TBN algorithm uses stochastically generated noise when moving the particles. Additionally, the simulations add Gaussian white noise to the true movements of the vehicles and the vehicles’ sensor measurements. To determine the relative average joint error, every simulation was run 100 times with the AUVs following the same paths.

#### 3.2.2. Field Trial AUVs

Field trials were conducted on Foster Reservoir near Sweet Home, Oregon, to validate the aforementioned techniques using real altimeter and acoustic communication data. Two micro-unmanned underwater vehicles (uUUVs) built by Riptide Autonomous Solutions (https://www.baesystems.com/en-us/product/riptide-family-of-autonomous-undersea-vehicles accessed on 1 January 2021) (see Figure 3) were used to collect data for the filed trials. Each uUUV is equipped with an acoustic modem, altimeter, IMU, and GPS receiver. A detailed list of the vehicles’ relevant hardware is presented in Table 1. The two uUUVs, known as Dory and Nemo, collected altimeter, acoustic communication, dead reckoning and GPS data while following dead reckoning paths on the surface of the reservoir. Using dead reckoning in this manner provided underwater-grade path following while allowing the vehicles to collect GPS data for ground truth comparisons.

The uUUVs’ proprietary software utilizes the Mission Oriented Operating System (MOOS). A MOOS app was designed to perform back and forth communication where each vehicle transmits an acoustic message to the other vehicle immediately upon receiving a communication. To begin the pattern, and in the event of a missed communication, Dory was programmed to transmit a message once every minute in the absence of a response. Nemo only transmitted a message upon receiving one from Dory. A one minute timeout was chosen because it takes about 30 s for a vehicle to process a received message and then queue up and send a response message.

The acoustic messages consisted of a message ID number and a character flag to indicate if the message was sent as a response to a received message, or if it had been initiated due to a timeout. One way time of flight (OWTOF) information was determined from the timestamps appended to each message when transmitted by the acoustic modem. To make the timestamps accurate enough for intervehicle ranging, the clocks on the uUUVs were synchronized via the network time protocol (NTP) with Dory acting as a server and Nemo as a subscriber. The NTP service was provided over wifi which was only available near the deployment site. Once the uUUVs departed on their paths the service was no longer available. The uUUVs then maintained time synchronization with their ChronoDot high accuracy real-time clocks. This provided OWTOF measurements with a standard deviation of about 10 milliseconds which equates to a standard deviation in ranging measurements of 14.75 m. This accuracy is considered sufficient for the purpose of this experiment given that the dead reckoning error of these vehicles often exceeds 100 m.

Nemo was found to have an erroneous compass that returns bad heading measurements when facing the southeast quadrant (90 to 180 deg.). Nemo’s paths initially compensated for this issue by subtracting 50 degrees off of headings that were originally between 90 and 180 degrees. Once GPS data were available from the experiment, the vehicle’s true headings were determined and a quadratic polynomial curve fit was used to correct the compass measurements. This technique was applied to both vehicles but its benefits are not as pronounced on Dory. The GPS data were also used to determine constant offsets to correct each vehicle’s speed. Figure 4 illustrates Nemo’s corrected and uncorrected dead reckoning paths. The green line is Nemo’s ground truth GPS path, and the red and blue lines are its dead reckoning and corrected dead reckoning paths, respectively.

#### 3.2.3. Vehicle Modeling

Each vehicle’s speed, compass, altimeter and acoustic modem OWTOF were modeled to improve the quality of the data and determine the uncertainties of the hardware. First, the vehicle’s GPS path was smoothed with a moving averaging filter. All of the vehicle trajectories should have a constant speed of two knots or 1.029 m per second. The vehicle’s true speed was taken to be the average of the speeds calculated from the distances between GPS locations and duration between the corresponding timestamps. The standard deviation of the speeds was set as the uncertainty in the vehicle’s true speed. The vehicle’s true headings for each time step were determined by the smoothed GPS data. The errors between the true headings and the corrected compass headings computed from the quadratic polynomial curve fit were calculated. The standard deviation of the errors was set as the uncertainty in the corrected compass readings. The motion of the vehicle can then be modeled with dead reckoning using a North-East-Down reference frame. See Equation (Equation 6).
(6)$$X_{t+1}=s_{t}\delta t_{t}\left[\begin{array}{c} sin(\theta_{t})\\ cos(\theta_{t}) \end{array}\right]+X_{t}$$
where
(7)$$X=\left[\begin{array}{c} x\\ y \end{array}\right]$$

The altimeter was modeled by first smoothing the data with a Kalman filter. The effect of the aforementioned outlier rejection and smoothing can be seen in Figure 5. Next, the smoothed GPS path was used to get a ground truth altimeter profile. The errors between the altimeter profile and the corresponding altimeter measurements were used to calculate the average altimeter error and its standard deviation. Similarly, the acoustic modem’s OWTOF was modeled by using the GPS data to determine the true distances between the vehicles for each received communication. Then, the mean and standard deviation of the errors between the true distances and the ranges calculated from the OWTOF, assuming a speed of sound in water of 1475 m per second, was calculated. Figure 6 depicts some of the ground truth data used for the modeling.

After the vehicle modeling was completed, the data from the selected mission were corrected based on the resulting vehicle models. For each vehicle, all nonzero speeds during the mission were set to the calculated true speed. The corrected compass headings were computed from the vehicle’s quadratic polynomial curve fit. The altimeter data were smoothed with a Kalman filter. The average altimeter error was then subtracted from all of the nonzero altimeter readings. The range measurements for the successful acoustic communications were calculated by subtracting the average OWTOF error from the successful communication measurements, then multiplying them by 1457 m per second. Table 2 illustrates the resulting vehicle models.

## 4. Results

This section shows the results from the simulations and field trials. The simulation results are presented first. This is followed by filed trial results which first, illustrates the bathymetry map that was produced of the experiment area, and then presents the communication bandwidth used by the planning algorithm and the resulting localization accuracy of dec-TBN.

### 4.1. Simulation

To the authors’ knowledge, previous dec-TBN works only use a full communication scheme where the AUVs take turns communicating at each time step. The planning algorithm results are compared to simulations with full communication and communication that happens on an incremented schedule. The comparison schedules varied the amount of bandwidth used. Lower bandwidth schedules involved the vehicles communicating in evenly spaced blocks. Each block of communication involved all of the vehicles taking a turn to communicate. The blocks were spaced so that the total number of communications used equaled a percentage of the full communication.

Results are presented for two sets of simulations. The first set involved two AUVs and the second set involved four AUVs. During each set of simulations, the AUVs localized with dead reckoning and dec-TBN, which was evaluated using the aforementioned communication regimes. Each set of simulations is an aggregation of 1100 individual simulations from 11 localization methods being evaluated 100 times each.

Table 3 and Table 4 list the localization methods used for the simulations. The bandwidths are listed in the left columns of the tables as part of the localization method. The left center columns list the corresponding number of communications. The right center columns show the total error for each communication scheme. The total error is calculated as the area under the curve of the relative average joint errors which can be seen in Figure 7, Figure 8, Figure 9 and Figure 10. The right columns list the average error which is calculated as the total error divided by the number of time steps. Note that dead reckoning error is not present in the figures because it is much greater than the errors being shown.

Figure 7 and Figure 9 show an overview of the joint errors for some of the communication policies evaluated in simulations with two and four AUVs, respectively. The presented policies include the policy produced by the planning algorithm, the full communication policy, some of the proportional communication policies, and non-cooperative TBN. Figure 8 and Figure 10 provide closer looks at the joint errors for the planned communication and lower bandwidth communication policies for simulations with two and four AUVs. Note that the 5% communication policy is present on all graphs for continuity.

Figure 7, Figure 8, Figure 9 and Figure 10 show that the proposed algorithm finds a communication schedule that provides more accurate localization. In all graphs the joint error experienced by the planned communication policy is less than the other communication policies. For the simulations with two AUVs the planning algorithm schedules a total of 67 communications which is similar to low bandwidth policies of 5% and 2.5% which use 50 and 26 communications, respectively. For the simulations with four AUVs the algorithm schedules 139 communications which is similar to the 10% bandwidth policy. This is nearly twice as many communications as prescribed by the planning algorithm for the two AUV simulations and achieves notably more accurate localization. This is due to the greater amount of information that is available from the two additional AUVs.

The lower bandwidth communication schemes perform better in general because they reduce the number of noisy measurements being transmitted. Additionally, the higher communication schemes can cause the TBN particle filters to become overly confident in their state estimate. This over confidence can result in the filter diverging, especially in these scenarios where at least one AUV is traveling through an area with minimal terrain features.

### 4.2. Field Trials

This subsection present the results of the experiments conducted at Foster Reservoir near Sweet Home, Oregon with two AUVs. First, the bathymetry map create by surveying the experiment area is shown. Then, the communication bandwidth used by the planning algorithm and the resulting localization accuracy are presented.

#### 4.2.1. Bathymetry Map

Creating a bathymetry map required surveying the portion of the reservoir used for the experiment. The bulk of the survey was performed by the Riptides operating on the surface for GPS localization. A Platypus LLC Lutra autonomous surface vehicle (ASV) equipped with a Lowrance side scan sonar was also used to survey a portion of the reservoir near the deployment site; however, the Lutra is reliant on wifi communication with a base station so its range was limited.

The accumulated altimeter and GPS data were used to create a 2.5D digital elevation profile of the reservoir’s bathymetry. First, the riptide altimeter measurements were adjusted to account for the attitude of the vehicle by using its roll pitch and yaw angles to project altimeter measurements into the world frame. The Riptides’ altimeters are mounted to look forward by 20 degrees which was added to the vehicles’ pitch before doing the transform. The Lutra’s sonar is mounted facing downward and attitude data were not available so the transforms were not applied to that data. Next, the survey and experiments were conducted over four days that were spread out over the course of a month. During this time, the height of the reservoir lowered by 4.1 m. There is a USGS hydrology station on the dam [20] that measures the elevation of the water’s surface. The hydrology data were used to calculate the difference in surface height between each day, which was applied to the Riptide and Lutra data. Then, the data were checked for outliers by first eliminating measurements that were deeper than the maximum depth of the reservoir, followed by applying a two sigma low-pass filter. Finally, the digital elevation profile was created using a sliding window Kalman filter to interpolate the data points onto an x-y grid of Universal Transverse Mercator (utm) coordinates. The resulting bathymetry map is presented in Figure 11.

#### 4.2.2. Experiments

Four missions were performed on Foster Reservoir to evaluate the communication planning algorithm. Each mission was evaluated using non-cooperative TBN, dec-TBN using all of the available communications (dec-TBN-full), dec-TBN using the communications that have been selected by the planning algorithm (dec-TBN-planned), and randomly generated communication policies.

When the communication planning algorithm generated policies for the two vehicles, the planning horizon was set to be the duration of the mission divided by 30 s which was the Riptides’ maximum communication rate assuming no communications were missed. The missions were between 24 and 38 min long resulting in planning horizons of 48 to 76 time steps. The planning algorithm was given the planning horizon along with each vehicle’s preplanned course, true speeds, and starting GPS position with an assumed uncertainty of three meters. The planning algorithm was also given noise models for the vehicles that were informed by the empirical models but slightly inflated, which is standard practice for a particle filter [21]. The algorithm then generated communication policies for the vehicles. The communication policies were matched to the available communications via their timestamps. Since the time steps for the communication policies incremented in 30 s intervals starting from the start of the mission, the available communications were matched to the closest time step in the policy. If the policy indicated that a communication should be used, then it was included in the dec-TBN update, otherwise it was omitted. The full communication rows of Table 5 shows the number of successful communications received by each vehicle during the missions. The Policy rows show the number of communications planned by our algorithm and the number of those communications that were used to evaluate dec-TBN. The table’s horizon row indicates the total number of possible acoustic transmission during the mission.

A communication regime that generated policies randomly was also used to evaluate dec-TBN. This served to compare the communication planning algorithm to naive communication planning. The random communication regime generated policies that utilized various amounts of bandwidth ranging from 80% to 5% of the vehicle’s potential communications, i.e., the vehicle’s mission duration divided by communication rate. These policies were created by placing the indicated number of communications randomly throughout the policy. The number of successful acoustic communications between the vehicles are limited; subsequently, the random communication policies are likely to miss the available communications. For this reason, when dec-TBN was evaluated with the random communication regime, each bandwidth was repeated 20 times with a new random policy generated for each iteration. Table 5 also lists the number of communications planned for the random policies, and the average number of planned communications that coincided with the available data.

The state estimation paths resulting from these communication regimes were compared to the vehicle’s GPS path. The errors and joint errors between the state estimations and the GPS path were calculated using Equations (Equation 4) and (Equation 5), respectively. The total errors for the two vehicles were then calculated as the Riemann sum of the joint errors, and the average errors were calculated as the total error divided by the duration of the mission.

Table 6 lists the total errors, average errors, and duration of each mission. Figure 12, Figure 13, Figure 14 and Figure 15 show the joint errors for each mission including the joint error from the corrected dead reckoning path, and the random communication policy that produced the lowest total error. The total errors are listed in the figures’ legends after the corresponding label. Images of the vehicles’ paths from all four missions are available in the appendix. These figures show the vehicle’s GPS path, corrected dead reckoning path, and the three TBN paths.

Table 5 and Table 6 shows that the planning algorithm used 43% or less of the available communication used by dec-TBN while maintaining or improving dec-TBN localization accuracy. Dec-TBN performed particularly well with the planning algorithm in missions 1 and 3. During mission 1, the communication policies utilized one communication from Nemo to Dory, whereas 12 communications were available between the vehicles. During the third mission, four communications were utilized from Dory to Nemo with a total of 10 communications being available. Figure 12 and Figure 14 also depict this trend with the planned communications joint error generally trending below the other errors.

Mission 2 had a relatively low error in corrected dead reckoning. Dec-TBN-planned is able to slightly improve the vehicles’ localization, while TBN and dec-TBN-full do slightly worse. It can be seen in Figure 16 that a portion of Dory’s TBN and dec-TBN paths follows a straight line that is parallel to the GPS and dead reckoning paths, but is offset by some distance. Particle filters are known to bounce around, so the straightness of this segment suggests that altimeter data were not available which caused the particle filter to track parallel to the dead reckoning path. The offset was likely caused by the particle filter diverging before encountering this gap in altimeter data. The poor performance of dec-TBN-full suggests that poor TBN localization during certain communications caused a negative feedback loop that degraded the dec-TBN-full state estimates. This can be seen in Figure 13 where the dec-TBN-full joint error spikes while the other errors stay low. The communication planning algorithm does not anticipate the quality of the data. In this case, the planned policy for this mission utilized five of the 21 available communications. This not only reduced the negative feedback, but also chose communications that were more informative.

In mission 4, dec-TBN-full achieves the lowest total error. The total error for dec-TBN-planned is about 21% larger than full communication, but it still performs better than TBN and dead reckoning while only using one of the 11 available communications for that mission. Additionally, the random communication policies are able to achieve similar localization results as the planning algorithm but they require a parameter sweep to identify the appropriate communications bandwidth. Subsequently, the communication planning algorithm is able to identify a competitive policy without trial and error.

## 5. Discussion

This paper has provided a communication planning algorithm for AUVs using decentralized TBN (dec-TBN). The AUVs are assumed to be equipped with an altimeter and an acoustic mode. The AUVs are also assumed to have knowledge of the other vehicle’s paths, basic motion and noise models, and their starting location and its uncertainty. Additionally, the AUVs are given a bathymetry map of the area they will be operating in. The communication planning problem is formulated as a constraint-based optimization. The problem is then solved by forward-simulating the AUVs’ dec-TBN algorithms along their intended paths and using the simulation states to construct a directed acyclic graph (DAG). After the simulation reaches its goal state, the communication policy is determined by doing a shortest path search through the DAG.

The communication planning algorithm was tested through simulations and field trials. The simulation leveraged a bathymetry map that had been created from exhaustive surveys of Lake Nighthorse, CO. It also utilized an altimeter model from an Ocean Server Ecomapper AUV. The simulations involved two or more AUVs traversing the bathymetry map along preplanned paths and localizing with dec-TBN. The dec-TBN was performed in accordance with a number of communication regimes, including communication policies generated by the planning algorithm, no communications, communications at every time step, and having communications spaced out at a variety of intervals. Each simulation was repeated 100 times. The effectiveness of the communication regimes was determined by the dec-TBN localization error calculated as the difference between the simulator’s location of the vehicles and the dec-TBN state estimates. The simulations showed that the communication planning algorithm provided a low bandwidth policy that also improved localization accuracy over the other communication regimes.

To validate the communication planning algorithm, field trials were conducted on Foster Reservoir, OR. Two Riptide micro-unmanned underwater vehicles (uUUVs) were used to collect data in the reservoir. The two uUUVs followed preplanned paths on the surface of the reservoir collecting GPS, altimeter, acoustic communication, and dead reckoning data. The data were then used to construct a bathymetry map of the area that the vehicles had been in. The data were also used to build instruments and process models of the uUUVs. The planning algorithm was used to generate communication policies for the paths followed by the uUUVs given their vehicle models and the bathymetry map.

The communication policies were evaluated by playing back the mission data and using a dec-TBN algorithm to localize the vehicles. Four communication regimes were used for this evaluation: no communication, full communication, the communication policy, and random communication policies of varying bandwidth. For full communication, all of the available acoustic communications were provided to the dec-TBN algorithm. For the communication policy, only the available acoustic communications that coincided with a planned communication were provided to the dec-TBN algorithm. Six random policies used varying numbers of communication placed at random time steps. The random communications were used if they coincided with an available acoustic communication.

The results show that the communication policy used 43% or less of the available communications and provided better localization on three of the four data sets. On the fourth data set, the full communication regime provided more accurate localization, but the communication policy maintained 86% of the localization accuracy while using 9% of the communications. The results from the field trials and simulations indicate that the policies produced by the planning algorithm reducing the communication bandwidth used for localization messages while maintaining, if not improving, the localization accuracy of dec-TBN.

## Figures and Tables

**Figure 1 sensors-21-01675-f001:**
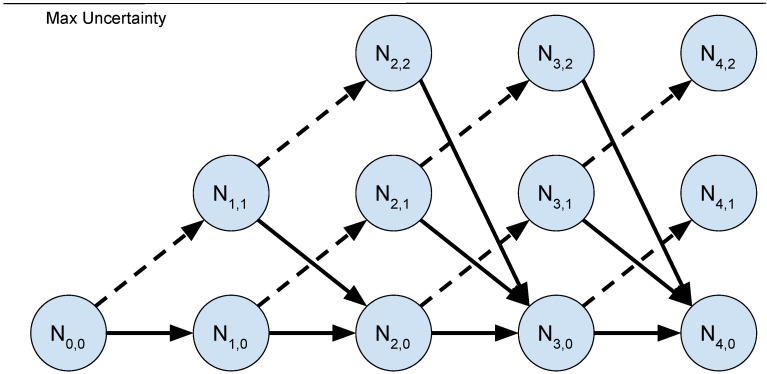
Communication planning graph. The node $N_{n}$ represents the *n*th node in *G*. Each node holds the value of σ of a simulation state. The dotted edges represent forward simulation without communication. The solid edges represent forward simulation with communication.

**Figure 2 sensors-21-01675-f002:**
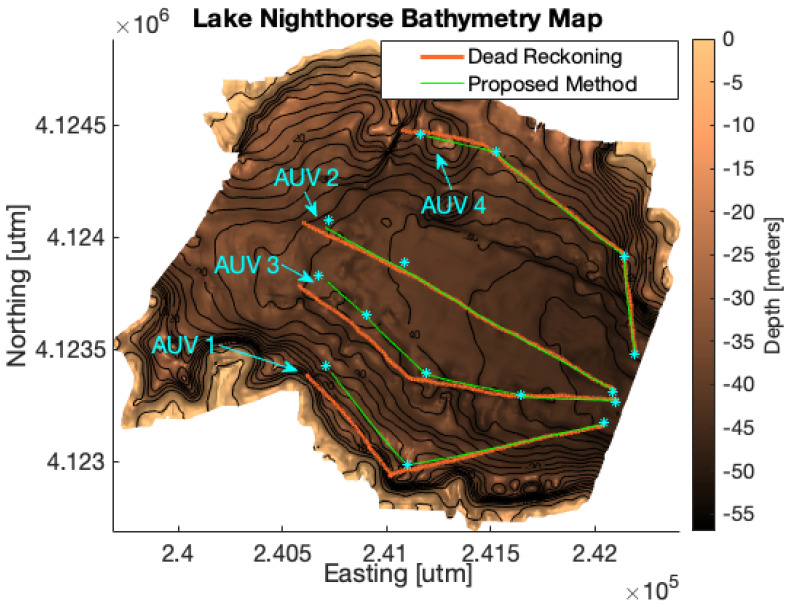
AUV paths through the simulated environment of Lake Nighthorse, CO. The blue stars are the AUVs’ waypoints. The green lines are the actual paths of the AUVs when using decentralized terrain-based navigation (dec-TBN), and the orange lines are the actual paths of the AUVs when using dead reckoning. Cooperative TBN leads to more accurate path following (closer to the waypoints) even in areas where the terrain is flat. The AUVs travel approximately 1.5 km.

**Figure 3 sensors-21-01675-f003:**
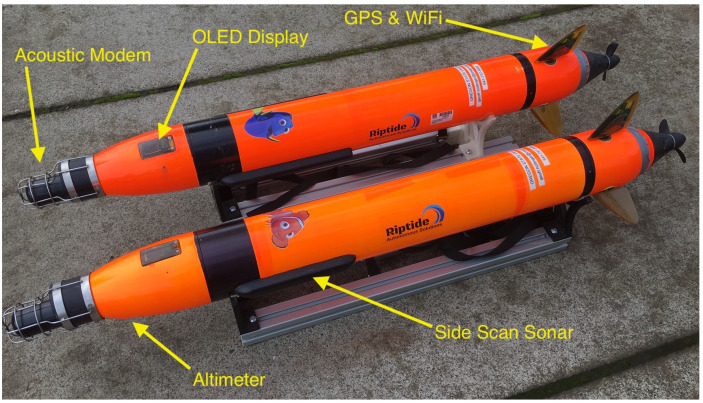
Riptide Autonomous Solutions micro-unmanned underwater vehicles (uUUVs) used for the experiments on Foster Reservoir, OR.

**Figure 4 sensors-21-01675-f004:**
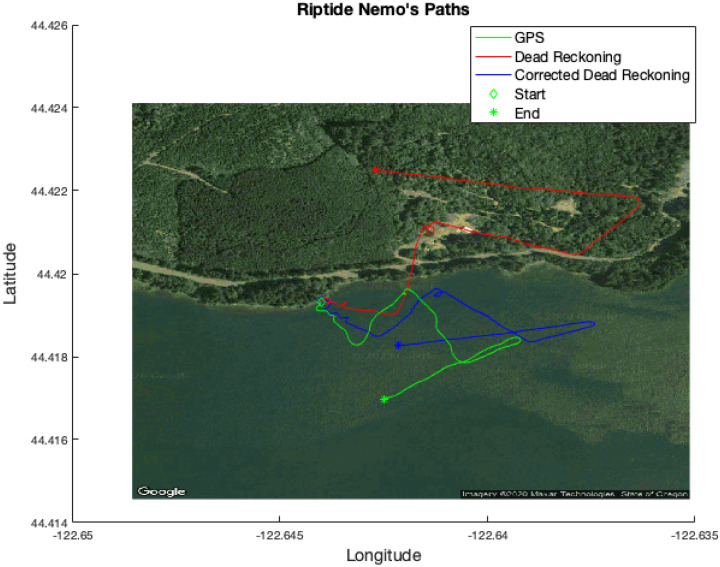
Correcting Nemo’s faulty compass. Ground truth GPS data (green line) were used to fit a quadratic polynomial curve to the faulty compass measurements that resulted in poor dead reckoning (red line). The GPS data were also used to determine a constant offset for the vehicle’s speed. This resulted in improved dead reckoning performance (blue line).

**Figure 5 sensors-21-01675-f005:**
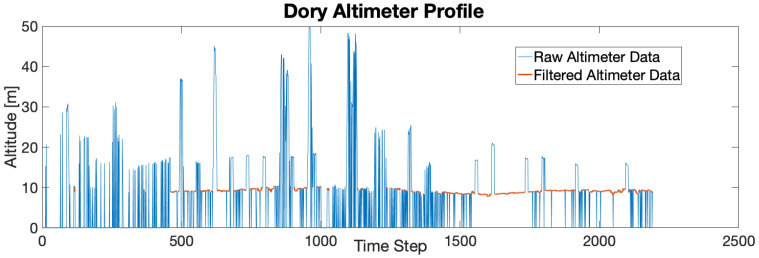
Dory’s altimeter profile from Mission 4. The blue line represents the raw altimeter data. The red line shows the filtered data after outlier rejection and Kalman filtering. Note that the reservoir is believed to have a maximum depth of 16 m.

**Figure 6 sensors-21-01675-f006:**
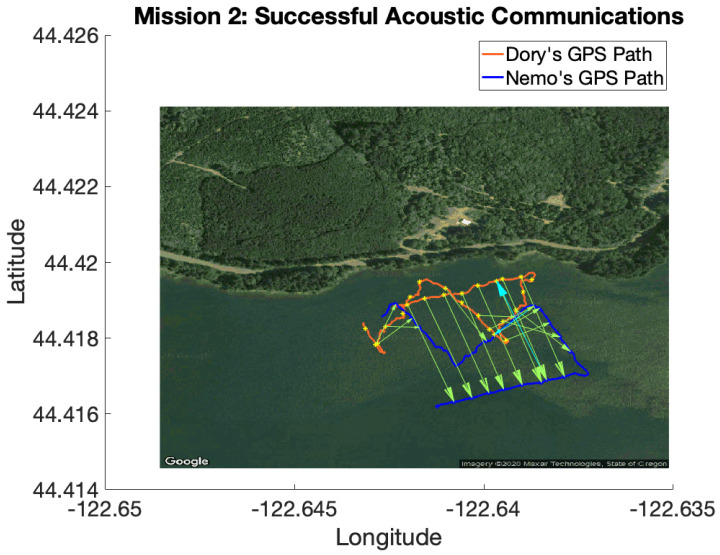
Successful acoustic communications between the uUUVs. The green arows indicate communications from Dory to Nemo and the Cyan arrows show communications from Nemo to Dory. Note that Dory was much more successful at communicating to Nemo.

**Figure 7 sensors-21-01675-f007:**
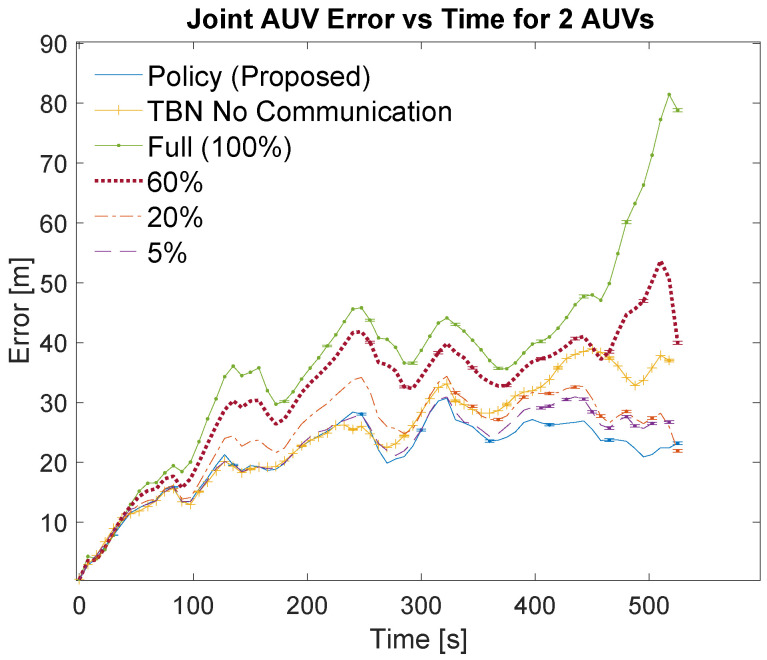
Overview of simulation results for 2 AUVs using varying communication bandwidths. Error bars show standard error of the mean.

**Figure 8 sensors-21-01675-f008:**
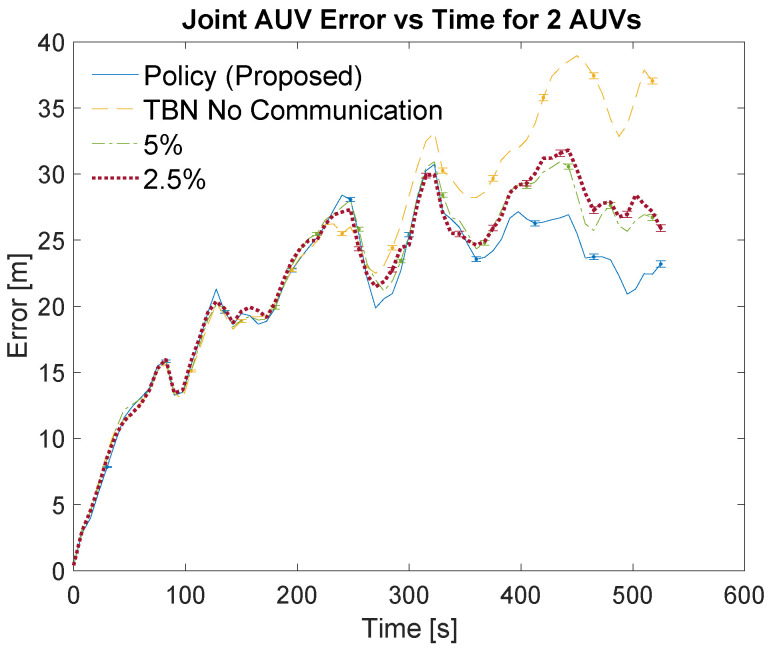
Simulation results for 2 AUVs using the planned policy, TBN without communication, and low bandwidth communication policies. Error bars show standard error of the mean.

**Figure 9 sensors-21-01675-f009:**
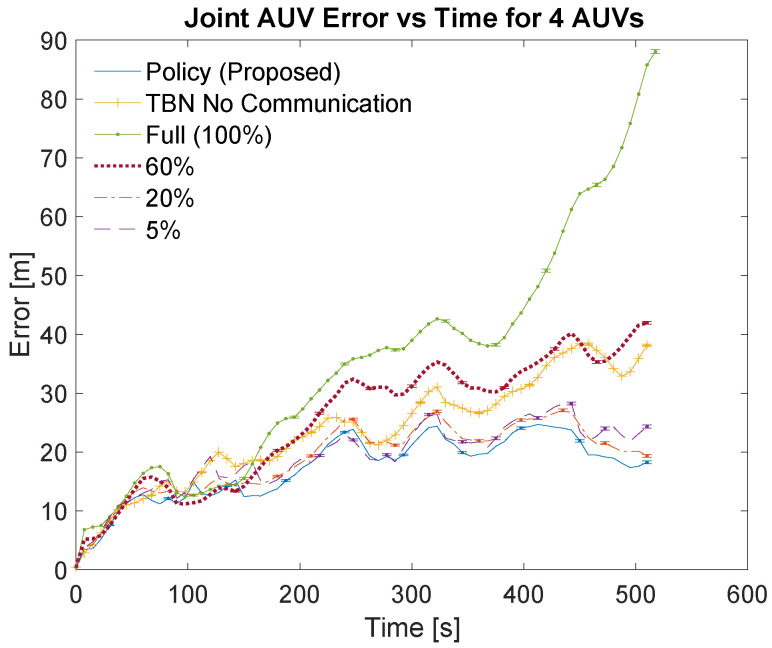
Overview of simulation results for 4 AUVs using varying communication bandwidths. Error bars show standard error of the mean.

**Figure 10 sensors-21-01675-f010:**
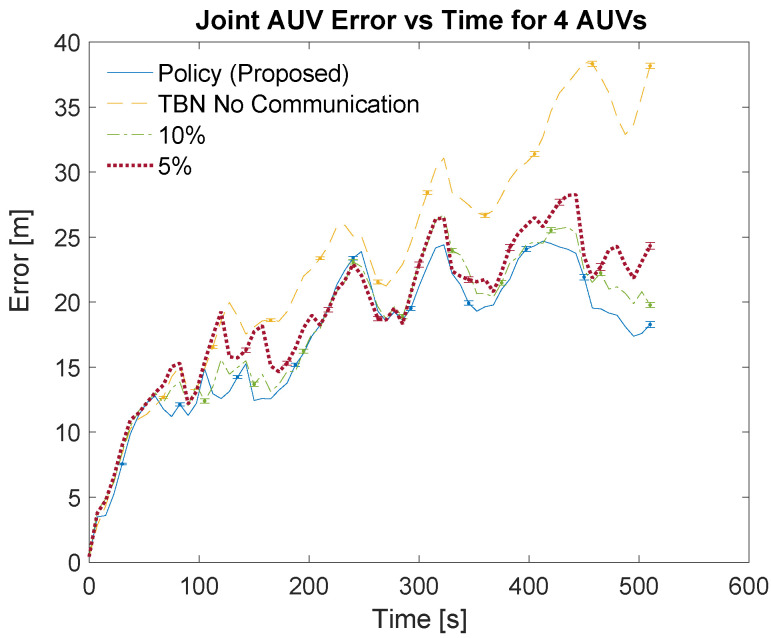
Simulation results for 4 AUVs using the planned policy, TBN without communication, and low bandwidth communication policies. Error bars show standard error of the mean.

**Figure 11 sensors-21-01675-f011:**
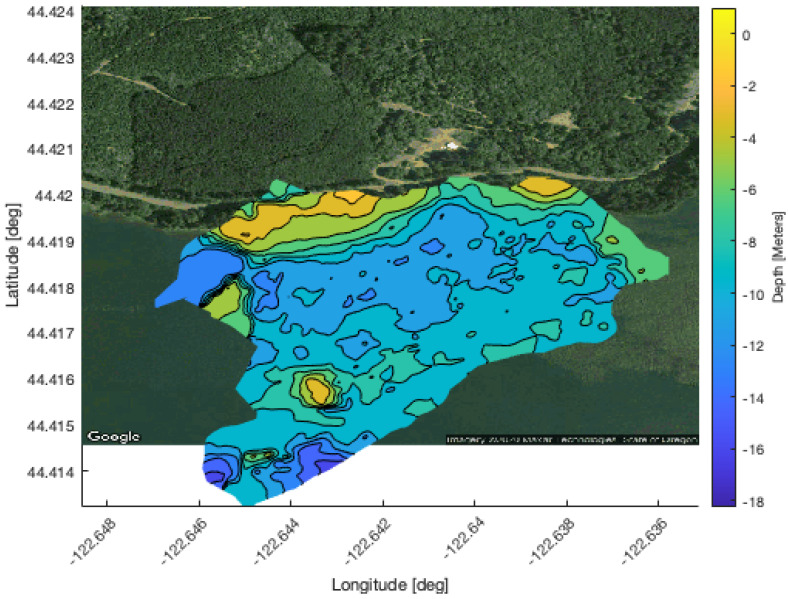
Bathymetry map of Foster Reservoir used for communication planning in decentralized terrain-based navigation field trials. This map a generated from data collected by the uUUVs.

**Figure 12 sensors-21-01675-f012:**
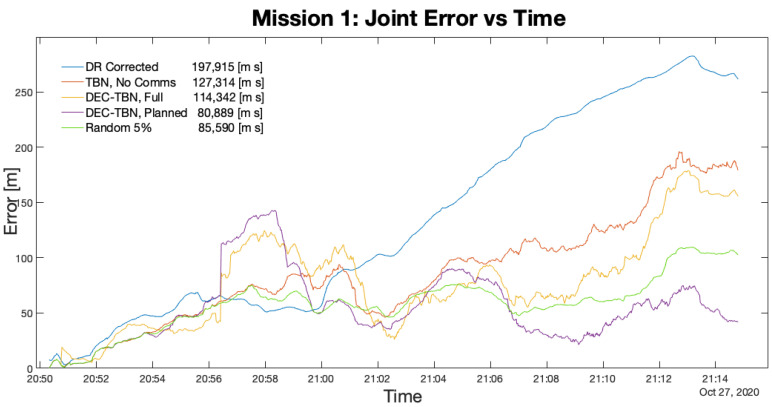
Joint Error for Mission 1. This figure shows that the planning algorithm enabled better localization accuracy. Only one communication was used by the policy.

**Figure 13 sensors-21-01675-f013:**
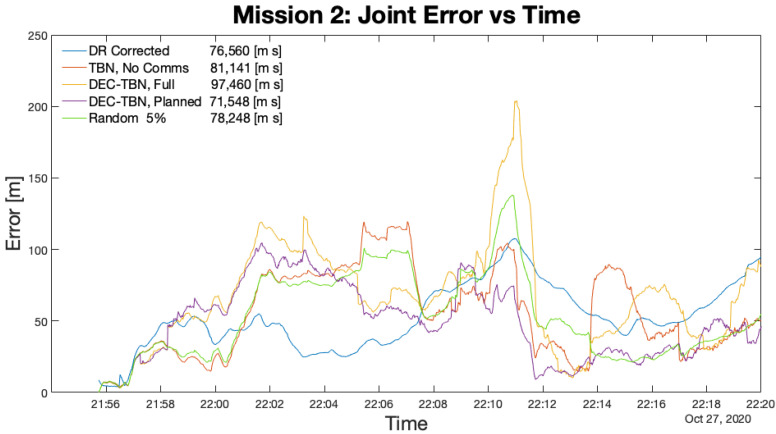
Joint Error for Mission 2. The spike in DEC-TBN, Full Comms. at 22:10 is indicative of a negative feedback loop from communicating erroneous localization data.

**Figure 14 sensors-21-01675-f014:**
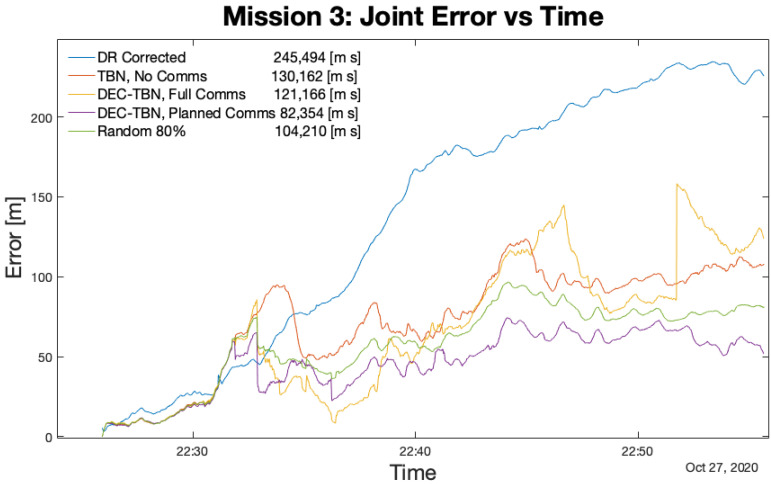
Joint Error for Mission 3. The planning algorithm produced the lowest total error using four communications. It was followed closely by a random policy that used six communications. In this case, the 80% random communication regime produced the lowest total error of the random communication regimes.

**Figure 15 sensors-21-01675-f015:**
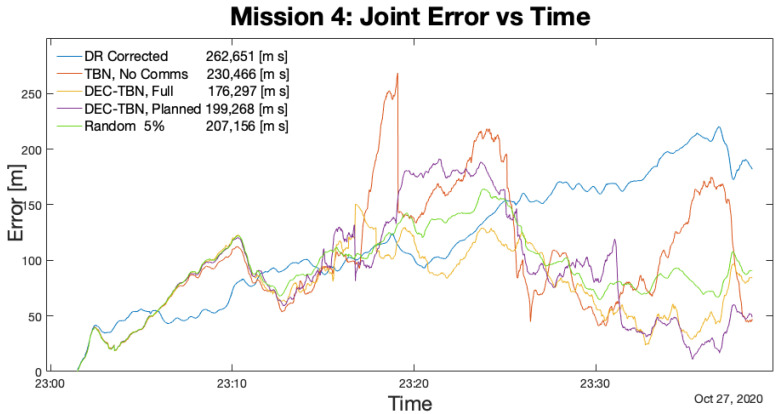
Joint Error for Mission 4. The spikes in TBN result from poor altimeter data for this mission. Dec-TBN full was able to compensate for the poor data better than dec-TBN planned. A low bandwidth random policy also happened upon a good set of communications for the circumstances.

**Figure 16 sensors-21-01675-f016:**
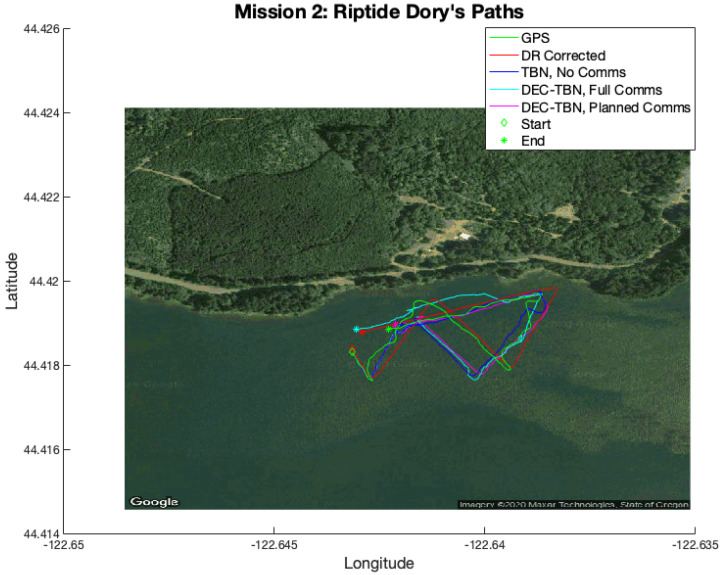
Dory’s paths from Mission 2. A gap in altimeter data is responsible for the straight section of the TBN paths that parallel the GPS and dead reckoning paths.

**Table 1 sensors-21-01675-t001:** Relevant Riptide Hardware.

Item	Part Name	Producer
Acoustic Modem	Micromodem-2	Woods Hole Oceanographic Institution
Altimeter	Sea Scan Echo	Marine Sonic Technology
GPS	Ivory3	GlobalTop Technology Inc.
IMU	AHRMS-M1	Sparton Navigation and Exploration
Real-Time Clock	ChronoDot	Macetech LLC
Computer	Beaglebone Black	Beaglebone

**Table 2 sensors-21-01675-t002:** Empirical Riptide Models.

	Process	Model	Units
Dory	Altimeter	−0.049 ± 1.774	m
	Acoustic Modem	0.653 ± 0.009583	s
	Speed	0.819 ± 0.249	m/s
	Compass	−2.277 ± 1.525	deg
Nemo	Altimeter	−0.034 ± 0.953	m
	Acoustic Modem	−0.652 ± 0.005731	s
	Speed	0.760 ± 0.201	m/s
	Compass	0.157 ± 23.784	deg

**Table 3 sensors-21-01675-t003:** Localization methods, communication bandwidth and total joint error for 2 AUVs over the course of a 1.5 km track.

Localization Method	Number of Communications	Total Error [m·s]	Average Error [m]
Policy (Proposed)	67	**11,004**	**10.53**
Dead Reckoning	0	73,644	70.4
TBN—No Comms.	0	12,911	12.36
dec-TBN, Full (100%)	1058	19,694	18.85
dec-TBN, 80%	816	17,645	16.89
dec-TBN, 60%	612	16,523	15.81
dec-TBN, 40%	408	15,088	14.44
dec-TBN, 20%	204	12,853	12.30
dec-TBN, 10%	102	12,044	11.52
dec-TBN, 5%	50	11,669	11.17
dec-TBN, 2.5%	**26**	11,782	11.27

**Table 4 sensors-21-01675-t004:** Localization methods, communication bandwidth and total joint error for 4 AUVs over the course of a 1.5 km track.

Localization Method	Number of Communications	Total Error [m·s]	Average Error [m]
Policy (Proposed)	139	**8929**	**8.54**
Dead Reckoning	0	73,514	70.3
TBN—No Comms.	0	12,180	11.66
dec-TBN, Full (100%)	1048	18,227	17.44
dec-TBN, 80%	840	15,489	14.82
dec-TBN, 60%	621	13,052	12.49
dec-TBN, 40%	416	11,066	10.59
dec-TBN, 20%	208	9730	9.31
dec-TBN, 10%	108	9341	8.93
dec-TBN, 5%	52	9820	9.39
dec-TBN, 2.5%	**28**	10,607	10.15

**Table 5 sensors-21-01675-t005:** Acoustic Communications

	Mission 1		Mission 2		Mission 3		Mission 4	
Horizon	48		48		59		74	
Dory	Plan	Used	Plan	Used	Plan	Used	Plan	Used
Full		7		2		3		5
Policy	8	1	7	0	7	0	12	1
Rand 80%	39	1.3	37	1.5	39	1.45	58	3.9
Rand 60%	29	1.05	28	1.3	29	1.65	43	2.75
Rand 40%	19	1.05	18	1.2	19	1.1	29	2.25
Rand 20%	9	0.55	9	0.4	9	0.6	14	1.25
Rand 10%	4	0.4	4	0.35	4	0.2	7	0.65
Rand 5%	2	0.15	2	0.2	2	0.1	3	0.45
Nemo								
Full		5		19		7		6
Planned	3	0	11	5	7	4	13	0
Rand 80%	24	3.25	37	13.45	24	4.65	51	3.5
Rand 60%	18	3.2	28	12.15	18	4.4	38	3.15
Rand 40%	12	1.7	18	9.85	12	3.45	25	2.05
Rand 20%	6	1.1	9	5.5	6	1.9	12	1.2
Rand 10%	3	0.6	4	3.1	3	0.6	6	0.8
Rand 5%	1	0.5	2	1.1	1	0.25	3	0.1

**Table 6 sensors-21-01675-t006:** Localization results from Foster Reservoir field trials. Path lengths range from 1.1 to 1.6 km.

Total Error	Mission 1	Mission 2	Mission 3	Mission 4	Units
DR Corrected	197,915	76,560	245,494	262,651	m·s
TBN, No Comms	127,314	81,141	130,162	230,466	m·s
DEC-TBN, Full	114,342	97,460	121,166	**176,297**	m·s
DEC-TBN, Planned	**80,889**	**71,548**	**82,354**	199,268	m·s
Random 80%	105,468	100,555	104,210	213,721	m·s
Random 60%	97,612	104,208	108,056	222,819	m·s
Random 40%	97,874	87,875	111,720	233,036	m·s
Random 20%	94,140	82,908	115,417	216,886	m·s
Random 10%	92,838	78,759	130,724	215,278	m·s
Random 5%	85,590	78,248	128,043	207,156	m·s
**Average Error**					
DR Corrected	135	52.5	137.6	117.8	m
TBN, No Comms	87	55.7	72.9	103.4	m
DEC-TBN, Full	78	66.9	67.9	**79.1**	m
DEC-TBN, Planned	**55**	**49.1**	**46.1**	89.4	m
Random 80%	71.8	69.0	58.4	95.9	m
Random 60%	66.4	71.5	60.6	100.0	m
Random 40%	66.6	60.3	62.6	104.6	m
Random 20%	64.1	56.9	64.7	97.3	m
Random 10%	63.2	54.1	73.3	96.6	m
Random 5%	58.3	53.7	71.8	93.0	m
**Duration**	24.5	26.25	29.7	37.1	min

## Data Availability

Not applicable.

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
