# Peer review of "Communication Planning for Cooperative Terrain-Based Underwater Localization"

_sensors, 2021, doi:10.3390/s21051675_

Round 1
Reviewer 1 Report
Recently, a lot of research is being conducted on the clustering algorithm of the unmanned marine vehicle. This paper is fascinating, and I think we can further refine the algorithm through more advanced experiments in the future.
There are vague parts in the text and need to be corrected.
Author Response
The paper has been fully revised for and edited for clarity
Reviewer 2 Report
- There are no mathematical equations dynamics to describe the autonomous underwater vehicles.
- More over, I encourage the authors to provide a more thorough discussion of the presented results. Finally, the formatting issues of this paper are really serious and sold be resolved.
Author Response
Vehicle kinematics have been added to section 3.2.3 Vehicle Modeling.
The paper has been fully revised and edited for clarity and formatting. The formatting for this journal is new to me and I believe that the paper is formatted correctly. If there are any remaining issues please be specific about them.
Reviewer 3 Report
I think it is an interesting methodology to position vehicles for certain vehicle applications. A high number of vehicles, therefore, requires a low cost in instruments and working in a known area.
Although the discussion indicates that the method used reduces the bandwidth by 43%, I think it would be interesting to indicate the ratio that the bandwidth is reduced compared to the dec-TBN, 2.5% since the tables indicate a comunication number more reduced while the error does not increase too much with respect to Policy (Proposed).
I think that the simulation times of the error graphs could be increased to verify stabilization of the errors.
The work proposes the calculation of the distance between the vehicles using the Time of Flight that requires a great synchronization in the clocks of the vehicles. The Time Difference of Arrival (TDOA)- could be used with two-way communications.
Author Response
The reported value of 43% is the ratio the bandwidth is reduced relative to Dec-TBN, so we believe that the reviewer's comment is already addressed in the paper. We have edited the text to make this clearer.
The simulations ended once the first vehicle reached its last waypoint. At this point the vehicle would likely surface and obtain a gps fix further improving localization accuracy.
TDOA is a feasible method for ranging two vehicles assuming the acoustic modems can respond in a manner that has a negligible latency. This method may not scale well to larger groups of AUVs since the vehicle transmitting its localization information would then have to take turns pinging each of the receiving vehicles. Though TOF requires precise timing, it only requires the transmitting vehicle to broadcast its information once and all of the AUVs in range will be able to receive that information if given proper modem addresses.
Reviewer 4 Report
This paper presents a decentralized communication planning algorithm for cooperative terrain-based navigation with autonomous underwater vehicles. The method is tested through simulations and a series of field trials. The results show that the algorithm can improve localization accuracy. Conclusions are well-supported by the experiments and give a clear summary of the results presented in the manuscript.
Author Response
Thank you